# PI4P-Containing Vesicles from Golgi Contribute to Mitochondrial Division by Coordinating with Polymerized Actin

**DOI:** 10.3390/ijms24076593

**Published:** 2023-04-01

**Authors:** Xinxin Duan, Yunfei Wei, Meng Zhang, Wenting Zhang, Yue Huang, Yu-Hui Zhang

**Affiliations:** Britton Chance Center for Biomedical Photonics—MoE Key Laboratory for Biomedical Photonics, Advanced Biomedical Imaging Facility—Wuhan National Laboratory for Optoelectronics, Huazhong University of Science and Technology, Wuhan 430074, China; duanxx2020@hust.edu.cn (X.D.);

**Keywords:** mitochondrial division, PI4P-containing vesicle, actin polymerization, interaction, super-resolution imaging

## Abstract

Golgi-derived PI4P-containing vesicles play important roles in mitochondrial division, which is essential for maintaining cellular homeostasis. However, the mechanism of the PI4P-containing vesicle effect on mitochondrial division is unclear. Here, we found that actin appeared to polymerize at the contact site between PI4P-containing vesicles and mitochondria, causing mitochondrial division. Increasing the content of PI4P derived from the Golgi apparatus increased actin polymerization and reduced the length of the mitochondria, suggesting that actin polymerization through PI4P-containing vesicles is involved in PI4P vesicle-related mitochondrial division. Collectively, our results support a model in which PI4P-containing vesicles derived from the Golgi apparatus cooperate with actin filaments to participate in mitochondrial division by contributing to actin polymerization, which regulates mitochondrial dynamics. This study enriches the understanding of the pathways that regulate mitochondrial division and provides new insight into mitochondrial dynamics.

## 1. Introduction

Lipid molecules are important components in cells that continuously carry out various dynamic processes to coordinate various important life activities [1,2]. Previous studies indicated that lipid molecules mainly maintain the stability of membrane lipids, regulate ion channels on the membrane, and facilitate cargo transportation [3,4]. However, it has recently been shown that lipid molecules play important roles in regulating the dynamic processes of organelles, such as division and polymerization [5]. For example, phosphatidylinositol-3-phosphate (PI3P) in Golgi-derived vesicles promotes endosomal division by regulating the content of PI3P at the endosomal division site [6], enriching our understanding of lipid functions.

Mitochondria participate in a series of biological processes and provide energy for cellular life activities, making them essential for cellular homeostasis, and they are genetically associated with multiple human diseases [7,8]. Mitochondrial fission and fusion create a functional mitochondrial network that is necessary for the distribution and homeostasis of mitochondria [7,9]. Mitochondrial division is necessary to regulate mitochondrial quality control and immune regulation [10], and numerous diseases are caused by mutations in genes encoding mitochondrial fission factors stressing the need to elucidate the mechanisms that regulate mitochondrial division [11]. Mitochondrial division is mainly regulated by many proteins, such as DRP1 and INF2 [12,13] and organelles, such as the endoplasmic reticulum (ER) and microfilaments [14,15]. However, recent research suggests that Golgi-derived PI4P-containing vesicles, which are lipid molecules, also play important roles in mitochondrial division. For example, it has been shown that Golgi-derived vesicles enriched with PI4P appear at the mitochondrial division site and act downstream of Drp1 [16]. In previous studies, PI4Ps were suggested to be involved in cellular regulation as signaling molecules [17,18]. However, the mechanism by which PI4P-containging vesicles affect mitochondrial division is unknown. Whether they participate in mitochondrial division in coordination with other organelles, such as the ER and microfilaments, is also unclear. Herein, using super-resolution microscopy, first, we show that PI4P-containing vesicles affect mitochondrial morphology by changing the PI4P lipid level, suggesting the importance of PI4P in the mitochondrial division process, which is consistent with recent studies [16]. Second, our results indicate that the transport of PI4P-containing vesicles in cells is associated with microtubules and the ER, which is beneficial to the study of lipid function. We also show that PI4P-containing vesicles can regulate the level of actin polymerization in cells by changing the level of PI4P lipid derived from the Golgi apparatus, and actin appears to polymerize at the contact site between PI4P-containing vesicles and mitochondria, causing mitochondrial division. Increasing the level of PI4P derived from the Golgi apparatus increases actin polymerization and reduces the length of mitochondria, suggesting that actin polymerization mediated by PI4P-containing vesicles is required for PI4P vesicle-related mitochondrial division, and that the involvement of PI4P vesicle function in mitochondrial division requires coordination of other components required for division. Together, our results support a model in which PI4P-containing vesicles and actin synergistically act at the site of mitochondrial division to complete the mitochondrial division process. Our finding indicating that PI4P lipid vesicles coordinate with actin polymerization and participate in mitochondrial division to regulate to mitochondrial dynamics, enriches the understanding of the mechanism of mitochondrial division and provides new insight into mitochondrial dynamics.

## 2. Results

### 2.1. Golgi-Derived PI4P-Containing Vesicles Regulate Mitochondrial Division

Previous reports suggested that Golgi-derived PI4P-containing vesicles are present at the site of mitochondrial division in the late steps of mitochondrial division [16]; however, their role in the mitochondrial division is still unknown.

To understand this process, we imaged PI4P-containing vesicles from the Golgi apparatus (tagged with fluorescent protein-mCherry2-FAPP1) and mitochondria (stained with MitoTracker Deep Red) in U-2 OS cells using structure illumination microscopy (SIM) (Figure 1). Our results showed that 69% of PI4P-containing vesicles were diffused in the cytoplasm, 31% of PI4P-containing vesicles had distributed on mitochondria (Figure 1a,b), and 5% occupied at the site of mitochondrial division (Figure 1c; Appendix A). Furthermore, we used small interfering RNAs (siRNA) to suppress endogenous PI4KIIIβ, a mammalian genome-encoded PI4P kinase critical for the generation of the PI4P pool in the Golgi vesicles [5], in U-2 OS cells (PI4KIIIβ-silenced cells) to reduce the content of PI4P. Mitochondria in PI4KIIIβ-silenced cells, PI4KIIIβ-overexpressing cells, and normal cells were imaged, and the length was measured. The results showed that when the level of PI4P decreased, the length of mitochondria increased compared with that of normal mitochondria, a finding in agreement with a previous report [16], suggesting that mitochondrial division events are regulated by the level of intracellular PI4P (Figure 1d,e; Appendix A). In PI4KIIIβ-overexpressing cells, the content of PI4P increased and the mitochondria were fragmented, which further indicated that mitochondrial division was affected by changes to the content of PI4P. When the PI4P content was increased, the mitochondrial length decreased, and the mitochondrial dynamics were altered, and the opposite effects were found when the PI4P content was decreased, suggesting that PI4P-containing vesicles derived from the Golgi apparatus regulate mitochondrial morphology by changing the PI4P content.

### 2.2. ER and Microtubules Assist the Transport of PI4P-Containing Vesicles in Cells

PI4P-containing vesicles derived from the Golgi apparatus are distributed in the cytoplasm. Some PI4P-containing vesicles must be translocated to the mitochondria for mitochondrial division to occur. However, the mechanism by which PI4P-containing vesicles are transported in cells is still unclear.

To explore this issue, we imaged PI4P-containing vesicles (tagged with fluorescent protein-mCherry2-FAPP1) and microtubules, microfilaments, intermediate filaments, and the ER (tagged with 3×mEmerald-ensconsin, EGFP-Lifeact, Halo-keratin, and EGFP-Sec61β, respectively) in U-2 OS cells using SIM. We analyzed the movement of individual PI4P vesicles and found that PI4P-containing vesicles exhibit various motility models in cells. According to the movement distance, we categorized PI4P-containing vesicles into three classes: long-distance transport, short-distance transport, and in situ, which accounted for 69%, 19%, and 12% of the vesicles, respectively (Figure 2a–d). The results showed that PI4P-containing vesicles appeared randomly on intermediate filaments or microfilaments but did not move along them (Figure 2e,f), suggesting that the transport of PI4P-containing vesicles may be not associated with intermediate filaments or microfilaments. Approximately 62.5% of PI4P-containing vesicles moved along microtubules including those that transported long distances and short distances, and the proportions of vesicles at these two forms were 67% and 33% (n = 500 events from 10 cells), respectively (Figure 2g,i,j), suggesting that PI4P-containing vesicle motility is dependent on microtubules, and that microtubules are more involved in supporting PI4P-containing vesicles for long-range transport. A total of 37.5% of PI4P-containing vesicles were associated with the ER over time, and they moved along the ER including those that transported long distances and short distances; the proportions were 32% and 68%, respectively (Figure 2h,i,k). These results suggested that the ER may assist PI4P-containing vesicles in moving inside cells and that the ER was more closely associated with PI4P vesicle short-range transport. Taken together, the data indicated that the transport of PI4P-containing vesicles was not associated with intermediate filaments or microfilaments. However, microtubules and the ER were associated with the transport of PI4P-containing vesicles, including long-distance and short-distance transport; additionally, microtubules were more closely associated with the long-distance transport of PI4P-containing vesicles and the ER was more closely associated with the short-distance transport of PI4P-containing vesicles. However, the ER is organized along the microtubules which are providing tracks for the ER; the observed vesicles’ movement along the ER tubule could actually be along the microtubules, which needs to be explored further.

### 2.3. PI4P-Containing Vesicles Participate in Mitochondrial Division by Coordinating Actin Polymerization

Previous studies have shown that actin plays important roles in mitochondrial division, and that the transient assembly of actin contributes to mitochondrial division [19,20]. We found that actin polymerized in the presence of PI4P-containing vesicles (Figure 3a–c). To evaluate the frequency of these colocalization events, we quantified the events that PI4P vesicles are at the site of actin polymerization. Approximately 47.4% of all actin polymerization sites were marked by PI4P-vesicles (Figure 3d). Therefore, we investigated whether PI4P-containing vesicles derived from the Golgi apparatus cooperate with other subcellular structure, such as actin, to participate in mitochondrial division.

To test this guess, we performed time-lapse imaging of actin filaments (tagged with EGFP-Lifeact) in normal cells (without treatment), LatB [21] (actin monomer–sequestering drug latrunculin B)-treated cells, PI4KIIIβ-overexpressing cells, and LatB-treated/PI4KIIIβ-overexpressing cells, respectively, using SIM. The results showed that actin continuously polymerized and depolymerized and each polymerization event was accompanied by an instantaneous increase in fluorescence intensity (Figure 3e,f). There were approximately 5 polymerization events observed in normal cells during the imaging time, and the number of actin polymerization events increased to approximately 10 in the cells overexpressing PI4KIIIβ. Negligible actin polymerization events were observed in LatB-treated cells; after PI4KIIIβ was overexpressed, the number of actin polymerization events in LatB-treated cells recovered compared to that in normal cells (Figure 3g,h). These results indicated that polymerization events were associated with the level of PI4P as the concentration of PI4P increased, and the number of actin polymerization events also increased.

To investigate whether PI4P-containing vesicles and actin polymerization act in concert during mitochondrial division, we imaged mitochondria, PI4P-containing vesicles, and actin in U-2 OS cells, simultaneously. It has been shown that actin plays important roles in mitochondrial division (Figure 4a; Appendix A) and our results showed that PI4P-containing vesicles and actin polymerization colocalized at the site of mitochondrial division (Figure 4b). To evaluate the frequency of these events, we quantified the events that actin polymerized at the PI4P vesicle–mitochondria contact sites. Approximately 38.5% of all PI4P–mitochondria contacts were marked by actin polymerization.

Approximately 52.6% of PI4P-containing vesicles marked mitochondrial division events that were also marked by polymerization actin (Figure 4c,e). An example of actin polymerization and PI4P participating in mitochondrial division was shown in Figure 4c. These findings suggested that PI4P-containing vesicles and actin polymerization may synergistically contribute to mitochondrial division. To eliminate the potential impact of sex, we also employed ARPE-19 cells (derived from a 19-year-old male). Our results showed that PI4P vesicles and actin polymerization were also present at the mitochondrial division sites in ARPE-19 cells, consistent with the above findings in U-2 OS cells (Figure 4d,e). Furthermore, mitochondria were imaged in normal cells, LatB-treated cells, PI4KIIIβ-overexpressing cells, or LatB-treated/PI4KIIIβ-overexpressing cells, respectively, and the mitochondrial number and mean area were quantified. The results showed that the morphology of mitochondria was different in these cells (Figure 4f,g). Quantitative results showed that in PI4KIIIβ-overexpressing cells, the mitochondrial mean area was decreased and the mitochondrial number was increased compared with those in normal cells. If actin acted downstream of PI4P vesicles, alterations in actin activity should inhibit PI4KIIIβ-induced mitochondria division. Actin polymerization suppression by LatB treatment reversed the effect of PI4KIIIβ on mitochondria size (Figure 4h). These results further suggested that the PI4P vesicle-mediated mitochondrial division occurred through actin polymerization.

## 3. Discussion

Previous reports have shown that PI4P-containing vesicles are present at the site of mitochondrial division, but the role played by these vesicles in the highly coordinated steps of mitochondrial division remains unclear. As the morphology of mitochondria, microtubules, microfilaments, and the ER is heterogeneous, with ultrastructural features that are below the diffraction limit, we used SIM [22,23,24] to capture the dynamics of subcellular structure better. We found that PI4P-containing vesicles regulate mitochondrial division by changing the content of PI4P derived from Golgi apparatus which is consistent with recent studies. Furthermore, we found that PI4P-containing vesicles regulated actin polymerization events and coordinated with actin polymerization to complete mitochondrial division.

Mitochondrial fission is a complex process that requires many factors [25,26]. During mitochondrial fission, GTPase kinetics-related protein-1 (Drp1) is recruited to the endoplasmic reticulum (ER)-induced mitochondrial contraction site, where it drives fission. However, the elements required to complete mitochondrial division are not known. Recently, PI4P-containing vesicles from Golgi have been proposed to facilitate these final steps of mitochondrial division [16]. To date, stable PI4P pools have not been reported in mitochondria, although the presence of its precursor PI has been described [27]. Although it has been proposed that Golgi-derived PI4P vesicle recruitment is Drp1 dependent, it is not clear how Golgi-derived PI4P vesicles reach the mitochondrial fission site, and whether they also act in concert with other subcellular structures to cause mitochondrial division. Regarding the role of PI4P in mitochondrial fission, other reports have highlighted the importance of phosphatidylinositol in mitochondrial dynamics. Lysosomal lipid transfer protein ORP1L may transport PI4P from lysosomes to mitochondria, and inhibition of its transfer or depletion of PI4P at mitochondrial fission sites impairs fission [28]. In contrast to previous studies, we proposed a new mode of PI4P action that differs from ORP1L transport. We found that PI4P-containing vesicles from Golgi can act as a signaling molecule to recruit actin to mitochondrial fission sites, and then actin can polymerize at PI4P-associated mitochondrial division sites. Previous reports showed that the forces generated by the actin skeleton can result in cellular membrane deformation and scission [29,30,31], and the tension generated by actin polymerization also causes mitochondrial inner and outer membranes to break down [20]. Our experimental results suggested that PI4P vesicles were not the ultimate mitochondrial division regulator as mentioned in previous studies [16], and that it may be the force generated by subsequent actin polymerization that led to the final division of mitochondria. There is a report that endosomal PI4P recruits the WASH complex to sort endosomes, which binds to Arp2/3-complex-dependent actin nucleation, and it regulates the division and segregation of endosomal products [5]. Endosomes and mitochondria appear to have similar division mechanisms, such as, the involvement of Golgi-driven vesicles containing PI3P in endosomal division^6^. Therefore, we speculate that PI4P may be involved in the recruitment of actin nucleation effectors similar to Arp2/3 that complete the final division of the mitochondrial membrane. Which actin nucleation effectors play a role needs further study.

We found that PI4P vesicles participated in mitochondrial division by regulating actin polymerization and regulated intracellular mitochondrial dynamics in a variety of immortalized cell lines. To avoid the effect of gender, we performed experiments using both U-2 OS cell line from a 15-year-old girl and the ARPE-19 cell line from a 19-year-old boy, respectively. We obtained the same result—that PI4P vesicles promote mitochondrial division by affecting actin polymerization in these cell lines. Therefore, this suggests that the experimental results we obtained are somewhat general. However, it may have different results within the primary cell line, which needs to be further explored.

In previous studies, microtubules and microfilaments were critical for the transport of subcellular structures in cells [32,33]. For example, microfilaments support lysosomal short-range movement and microtubules can provide orbits through which lysosomes move rapidly at long distances in cells [34,35,36]. Although PI4P lipids are essential for the physiological functions of cells [17], little research has been carried out on the transport of PI4P-containing vesicles. Our results showed that the transport of PI4P-containing vesicles was associated with microtubules and the ER, microtubules were more closely associated with long-distance transport and the ER was more closely associated with the short-distance transport of PI4P-containing vesicles. Thus, the ER also plays an important role in the transport of PI4P-containing vesicles. The ER extends throughout the cell and occupies a large fraction of the cytoplasmic volume and may also exert an essential effect on the transportation of substances within the cell. There are also reports that the ER promotes local endosome translocation [37]. The ER enriches the intracellular material transport pathway.

In conclusion, our results demonstrate that PI4P vesicles from the Golgi apparatus are transported intracellularly along microtubules or ER and complete the final mitochondrial division by regulating actin polymerization. Our findings illustrate the importance of synergy between subcellular structures, which is consistent with other recent findings.

## 4. Methods

### 4.1. Cell Culture and Transfection

U-2 OS cells (human osteosarcoma cells) from a 15-year-old girl were cultured in 10% (*v*/*v*) FBS (fetal bovine serum; Gibco) and McCoy’s 5A medium (Gibco). ARPE-19 cells from a 19-year-old boy were cultured in 10% (*v*/*v*) FBS (fetal bovine serum; Gibco) and DMEM medium (Gibco). All the cell culture medium contained 1% antibiotic-antimycotic (Gibco) and cells were maintained at 37 °C with 5% CO_2_ during culturing. Approximately 1 × 10^5^ of U-2 OS cells per well were seeded into a 24-well plate. After 24 h, transfection was performed with OPTI-MEM medium (Invitrogen) and Lipofectamine LTX (Invitrogen) according to the standard protocol. Cells were digested with trypsin (Thermo Fisher Scientific) 5 h after transfection and seeded onto coverslips. Cells were imaged 24–48 h post-transfection and cultured at 37 °C with 5% CO_2_.

### 4.2. Latrunculin B (LatB) Treatment

Cells were seeded into 15 mm cell culture dishes (NEST) 24 h before the experiment and reached 70–90% confluency. Cells were transfected with EGFP-Lifeact plasmid the day before treatment and incubated with medium containing 0.5 μM latrunculin B for 1 h; DMSO was used as the negative control.

### 4.3. Plasmids

The EYFP-FAPP1 vector was kindly provided by Prof. Marino Zerial (Max Planck Institute of Molecular Cell Biology and Genetics, Dresden, Germany). The mCherry2 segment with BglII and EcoRI sites was inserted into the corresponding sites of the linearized EYFP to generate mCherry2-FAPP1. The EGFP-Sec61β vector was a gift from L. Chen (Peking University). The 3×mEmerald-ensconsin vector and Halo-keratin were generously provided by Prof. Dong Li (Institute of Biophysics, Chinese Academy of Sciences, Beijing, China). Lifeact-EGFP was artificially constructed based on the EGFP-N1 backbone, and mCherry2-PI4KIIIβ was artificially constructed based on the mCherry2-N1 backbone.

For all experiments, the following amounts of DNA were transfected per well: 300 ng with EGFP-Lifeact, 400 ng with mCherry2-FAPP1, 500 ng with Halo-keratin, 500 ng with EGFP-Sec61β, and 500 ng with 3×mEmerald-ensconsin.

### 4.4. Halo Probes

U-2 OS cells were seeded into 24-well plates (approximately 1 × 10^5^ cells per well) 15 h prior to the transfection experiment. The transfection of each plasmid was carried out using Lipofectamine LTX reagent with PLUS reagent (Invitrogen) according to the manufacturer’s instructions. Five hours later, the cells in each well were digested with trypsin-EDTA (0.25%) (Gibco), and the cells were seeded into 15 mm cell culture dishes at a density of 8 × 10^4^ cells per dish and cultured at 37 °C with 5% CO2 for 24 h. The cells were washed twice with PBS and then incubated with 5 μM Halo probes for 1 h at 37 °C with 5% CO_2_. The cells were washed twice with PBS and cultured for 1 h in culture medium (with 10% FBS) before imaging.

### 4.5. RNAi Transfection and Real-Time PCR

PI4KIIIβ was depleted using siRNA [38], 5′-GCACUGUGCCCAACUAUGA[dT][dT]-3′. Approximately 1 × 10^5^ of U-2 OS cells per well were plated into a 24-well plate. The cells were first transfected with 40 nM RNA oligonucleotides and 40 nM negative control siRNA using Lipofectamine^TM^ RNAiMAX (Thermo Fisher Scientific) in Gibco reduced-serum OPTI-MEM. After 6 h of RNAi transfection, the cells were washed and the medium was replaced with McCoy’s 5A medium supplemented with 10% FBS and 1% antibiotic-antimycotic. After 48 h, the cells were incubated with MitoTracker Deep Red. Total RNA was extracted from different siRNA-treated samples using TRIzol (Invitrogen) reagent, and cDNA was prepared by reverse transcription using a reverse transcription kit. The knockdown efficiency of different siRNAs was confirmed using SYBR Green Mix (Takara) with real-time PCR. The PCR primers were as follows:PI4KIIIβ forward primer, 5′-GAAGCGGGTGCCTATCTTGT-3′;PI4KIIIβ reverse primer, 5′-GGAACCAATCTTAGGAGGGAGT-3′.

### 4.6. Structured Illumination Microscopy (SIM) Imaging

SIM imaging was performed with a high-sensitivity structured illumination microscope (HiS-SIM, Guangzhou Computational Super-Resolution Biotech Co., Ltd., Guangzhou, China) or grazing incidence-structured illumination microscopy (GI-SIM) equipped with 488, 560, and 638 nm excitation lasers and a 100×/1.49-NA oil-immersion objective. All the superresolution images were reconstructed from 9 raw images. MATLAB and Fiji software were used to analyze the super-resolution images.

### 4.7. Determining the Optimal Condition to Image Intracellular Organelles

U-2 OS cells transiently transfected with PI4P vesicle markers and mitochondrial markers were imaged live by SIM every 2 s for 3 min. The relationship between mitochondria and PI4P-containing vesicles is shown in Figure 1a. Mitochondrion-associated PI4P-containing vesicles are defined as PI4P-containing vesicles in contact with mitochondria for more than 10 s.

The three-way contact made between the PI4P-containing vesicles, actin, and mitochondria was evaluated by quantifying the proportion of PI4P vesicle–mitochondria contacts at sites marked by actin, as determined with cells expressing PI4P vesicle markers (mCherry2-FAPP1) and actin markers (GFP-Lifeact). Random contacts in 10 cells were analyzed, and the contacts analyzed had formed before the video was recorded and lasted for at least 10 s.

The mitochondrial morphology was analyzed by ImageJ software. A region of interest (ROI) of 225 μm^2^ was selected, and the mean area and the number and length of mitochondria within the ROI were measured using the “Analyze Particles” plugin. Then, images were processed with the “smooth” function in ImageJ, manual thresholding was performed, and “Make Binary” and “Skeletonized” were applied.

### 4.8. Tracking PI4P-Containing Vesicles

Time-lapse images were acquired at 1.5 s intervals using a HiS-SIM. The trajectory of PI4P-containing vesicles was plotted using ImageJ software, and images were manually tracked by “Manual Tracking”. Then, GraphPad Prism software was used to generate motion trajectories according to the position coordinates. “In situ” was described as a trajectory < 1 μm, “short distance” was described as a trajectory < 10 μm, and “long distance” was described as a trajectory > 10 μm.

### 4.9. MitoTracker Deep Red Staining

Cells were seeded in 15 mm cell culture dishes (NEST) 24 h before the experiment and reached 70–90% confluency. Cells were incubated with 500 nM MitoTracker Deep Red FM for 30 min at 37 °C with 5% CO_2_ in regular medium, which was McCoy’s 5A medium with 10% FBS. Cells were washed with PBS and imaged.

### 4.10. Statistical Analysis

All statistical analyses and *p*-value determinations were performed in GraphPad Prism 8.

## Figures and Tables

**Figure 1 ijms-24-06593-f001:**
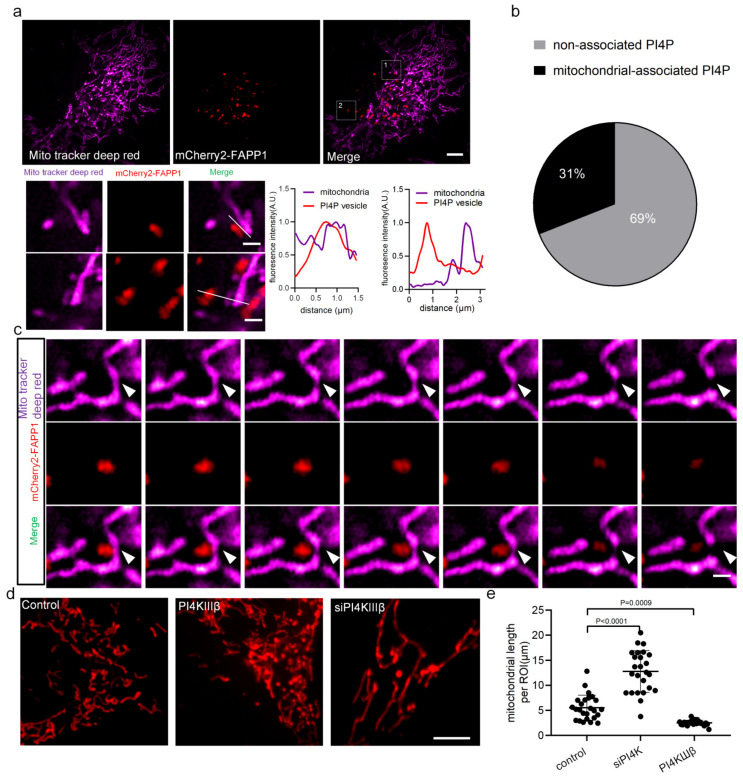
PI4P-containing vesicles regulate mitochondrial division. (**a**) SIM images of PI4P-containing vesicles (tagged with mCherry2-FAPP1, red) and mitochondria (stained with MitoTracker Deep Red, magenta) in U-2 OS live-cells. Two areas from **a** were shown at enlarged sizes with a line scan of the dotted line on the right Scale bar: 5 μm, and 1 μm. (**b**) Pie charts showing mitochondrion-associated versus -unassociated PI4P vesicles puncta. Our criterion for association with mitochondria was that PI4P-containing vesicles persist on mitochondria for at least 10 s. The percentage was calculated by determining the percentage of mitochondrial PI4P vesicles puncta at 3 min of imaging time in 300 PI4P-containing vesicles from 10 cells. (**c**) The images show a PI4P vesicle at the site of mitochondrial division. White arrows indicate the site of mitochondrial division. Scale bar: 2 μm. (**d**) SIM images showing mitochondrial morphology (stained with MitoTracker Deep Red, red) in U-2 OS cells treated with the indicated small interfering RNAs (siRNAs), Scale bar: 5 μm. (**e**) The quantification of mitochondrial length to determine the mean mitochondria length per region of interest (ROI) (225 μm^2^) in 20 cells. Cells from three independent experiments. All graphs show the mean ± SEM. Statistical significance: *p* < 0.001; *p* < 0.0001.

**Figure 2 ijms-24-06593-f002:**
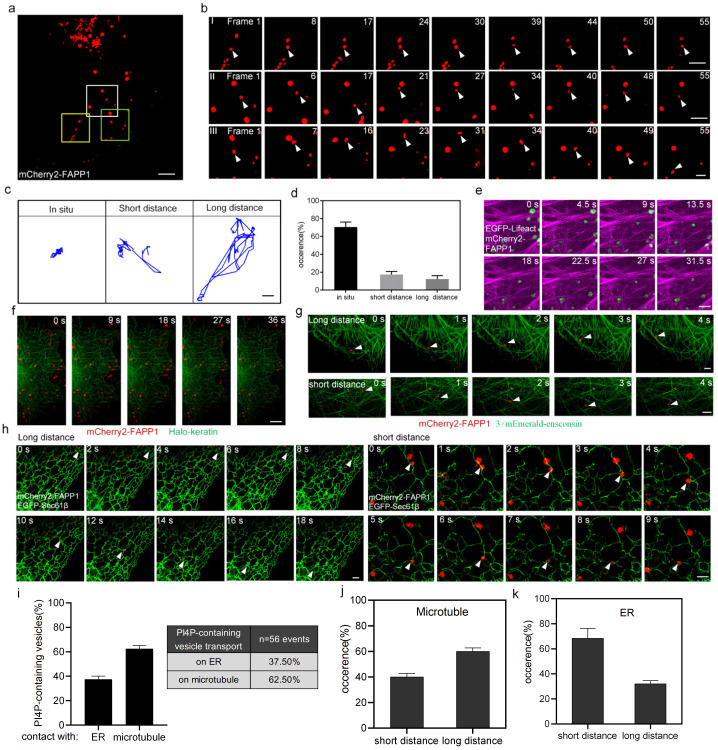
The ER and microtubules facilitate the intracellular transport of PI4P-containing vesicles. (**a**) SIM image of PI4P-containing vesicles (tagged with mCherry2-FAPP1, red) in live U-2 OS cells. Scale bar: 5 μm. (**b**) Boxed regions of interest shown in a are presented at the indicated time points to exemplify in situ, short-distance and long-distance transport. I is the region in the yellow box, II is the region in the white box, and III is the region in the green box. Scale bar: 2 μm. (**c**) Left panel, the trajectory of in situ PI4P vesicle transport; middle panel, the trajectory of short-distance PI4P vesicle transport; right panel, the trajectory of long-distance PI4P vesicle transport. (**d**) Histograms showing the percentage of the three kinds of PI4P-containing vesicle transport—in situ, short-distance and long-distance—as determined with 500 PI4P-containing vesicles in 10 cells. (**e**) SIM images showing PI4P-containing vesicles (tagged with mCherry2-FAPP1, green) and microfilaments (tagged with EGFP-Lifeact, magenta) in live U-2 OS cells. Scale bar: 2 μm. (**f**) SIM images of PI4P-containing vesicles (tagged with mCherry2-FAPP1, red) and intermediate filaments (tagged with Halo-keratin, green) in live U-2 OS cells. Scale bar: 5 μm. (**g**) SIM images of PI4P-containing vesicles (tagged with mCherry2-FAPP1, red) and microtubules (tagged with 3×mEmerald-ensconsin, green) in live U-2 OS cells. On the top, white arrows indicate the short-distance PI4P vesicle transport along microtubules; on the bottom, white arrows indicate the long-distance PI4P-containing vesicle transport along microtubules. Scale bar: 2 μm. (**h**) SIM images of PI4P-containing vesicles (tagged with mCherry2-FAPP1, red) and the ER (tagged with EGFP-Sec61β, green) in live U-2 OS cells. On the left, white arrows indicate the long-distance PI4P-containing vesicle transport along the ER; on the right, white arrows indicate short-distance PI4P vesicle transport along the ER. Scale bar: 2 μm. (**i**) Quantitative statistics of the relative localization of the PI4P-containing vesicles and the ER or microtubules and the transport events between the PI4P-containing vesicles and the ER or microtubules. (**j**,**k**) Histograms showing the percentage of short-distance and long-distance PI4P-containing vesicle transport along microtubules (on the left) and the ER (on the right).

**Figure 3 ijms-24-06593-f003:**
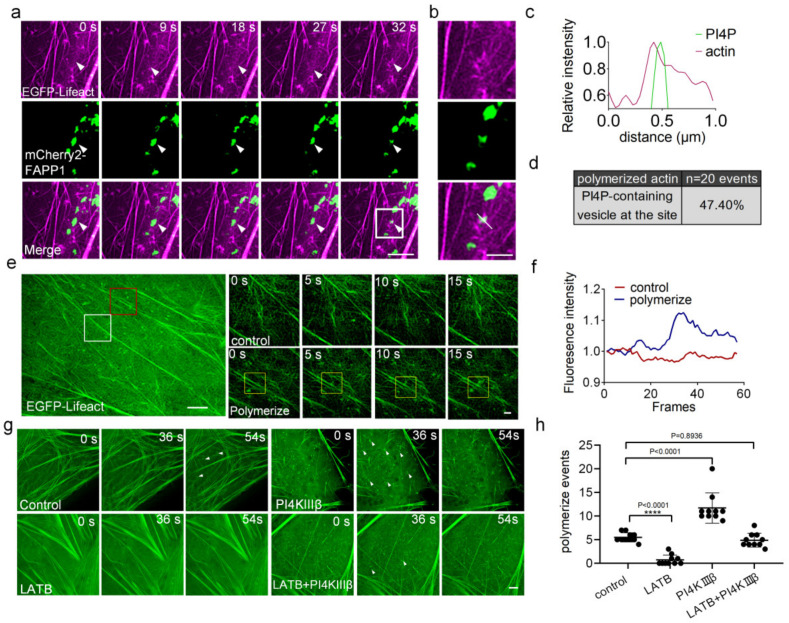
PI4Ps promote actin polymerization in cells. (**a**) Representative SIM images of PI4P-containing vesicles (tagged with mCherry2-FAPP1, green) and actin (tagged with EGFP-Lifeact, magenta) in live U-2 OS cells. Scale bar: 2 μm. (**b,c**) White arrows from (**a**) show actin filament polymerization where PI4P vesicles are present. Line-scan analysis of relative fluorescence intensities from the white line in the merged image is shown. Scale bar: 2 μm. (**d**) Quantitative statistics of the relative localization of the PI4P-containing vesicles and actin polymerization. (**e**) SIM images showing actin filaments (tagged with EGFP-Lifeact). The white box indicates a site without actin polymerization; on the top right panel, there is a long-term image showing actin dynamics. The red box indicates a site with actin polymerization; below the right panel, there is a long-term image showing actin polymerization. Scale bars: 5 μm and 1 μm. (**f**) Analysis of the relative fluorescence intensities in the red and white boxes is shown. (**g**) Time course analysis of changes in actin filaments in normal cells and cells in the presence of LatB or PI4KIIIβ or both LatB and PI4KIIIβ. White arrows indicate the site of actin polymerization. Scale bar: 2 μm. (**h**) Quantification of the number of actin polymerization events as presented in picture (**g**) on the basis of 10 cells. Cells from three independent experiments. All graphs show the mean ± SEM. Statistical significance: n.s., *p* > 0.05; **** *p* < 0.0001.

**Figure 4 ijms-24-06593-f004:**
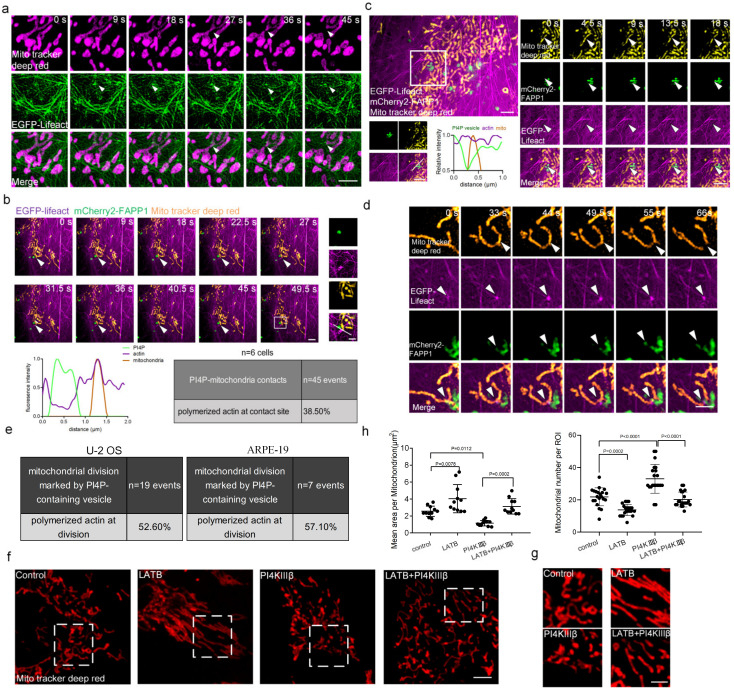
PI4P-containing vesicles participate in mitochondrial division by coordinating polymerized actin. (**a**) Representative SIM images of actin filaments (tagged with EGFP-Lifeact, green) and mitochondria (stained with MitoTracker Deep Red, magenta), showing that actin polymerized at the site of mitochondrial division. Scale bar: 2 μm. (**b**) Long-term SIM images of PI4P-containing vesicles (tagged with mCherry2-FAPP1, green), actin filaments (tagged with EGFP-Lifeact, magenta) and mitochondria (stained with MitoTracker Deep Red, yellow) in live U-2 OS cells. Inserts show the area indicated by the white box with a PI4P vesicle–mitochondrion contact marked by the actin polymerization at a higher magnification. Our criterion for identifying a three-way contact was the presence of actin at a PI4P vesicle–mitochondrion contact site for at least 10 s. Line-scan analysis of relative fluorescence intensities from the white line in the merge is shown. Percentage of PI4P vesicle–mitochondrion contacts marked by the actin polymerization in 15 cells. Scale bar: 2 μm and 1 μm (insert). (**c**) Representative SIM images of PI4P-containing vesicles (tagged with mCherry2-FAPP1, green), actin filaments (tagged with EGFP-Lifeact, magenta), and mitochondria (stained with MitoTracker Deep Red, yellow) in live U-2 OS cells, showing PI4P-containing vesicles and actin polymerization at the site of mitochondrial division; the right panel displays long-term images in the white box, with a line-scan analysis of relative fluorescence intensities from the white line in the merged image shown. Scale bar: 2 μm, 1 μm. (**d**) Representative SIM images of PI4P-containing vesicles, actin filaments, and mitochondria in live ARPE-19 cells. Scale bar: 1 μm. (**e**) Percentage of PI4P-containing vesicles marked wth mitochondrial division events that were also marked by polymerization actin in 25 U-2 OS cells (left) and in 5 ARPE-19 cells (right). (**f**) Representative images showing mitochondrial morphology in U-2 OS cells treated with LatB or PI4KIIIβ or both LatB and PI4KIIIβ or DMSO. Scale bar: 5 μm. (**g**) Four areas from **f** were shown at enlarged sizes, scale bar: 2 μm. (**h**) Quantification of mitochondrial morphology based on the images presented in (**f**) to determine the mean area per mitochondria in 10 cells and mitochondrial number per ROI in 20 cells. Cells from three independent experiments. All graphs show the mean ± SEM. Statistical significance: *p* < 0.05; *p* < 0.01; *p* < 0.001; *p* < 0.0001.

## Data Availability

Data is contained within the article and Appendix A.

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
