# Peer review of "PI4P-Containing Vesicles from Golgi Contribute to Mitochondrial Division by Coordinating with Polymerized Actin"

_ijms, 2023, doi:10.3390/ijms24076593_

Round 1
Reviewer 1 Report
IJMS
COMMENTS TO THE EDITORS AND THE AUTHORS
ijms-2288595: “PI4P-containing vesicles from Golgi contribute to mitochondrial division by coordinating with polymerized actin”
Dear the Editors and the Authors,
Please find enclosed the comments for the above-mentioned manuscript.
A SUMMARY OF THE CONTENT
The authors stated that the study reported the PI4P-containing vesicles derived from the Golgi apparatus regulated mitochondrial morphology by changing the content of PI4P and that actin appeared to polymerize at the contact site between PI4P-containing vesicles and mitochondria. Increasing the content of PI4P derived from the Golgi apparatus increased actin polymerization and reduced the length of the mitochondria.
THE OVERALL OPINION OF THE MANUSCRIPT
The strengths: the manuscript is within the scope of the journal and presents eventually interesting knowledge; the results were obtained using the different methods; the figures nicely present results.
The limitations: the idea is not completely new (https://pubmed.ncbi.nlm.nih.gov/?term=PI4P+division); the title and conclusions are not fully supported by the data; the descriptions and discussions are not precisely formulated; the original, and important pioneered results, as well as recent advance in the field focusing on the subject of the study are not cited in the introduction and they are not discussed; the experimental design is extremely artificial since study is based only on one specific immortalized cell line and physiological/biological meaning is missing; the intra- and inter- assay coefficients are not presented.
Accordingly, the new experiments/results and major revision are required for the future consideration.
Please find enclosed some of the suggestions in the comments to the authors listed below.
Dear the Editor and the Authors,
Please find enclosed some of the comments for the above-mentioned manuscript.
(1) TITLE
1.1. Please notice that the title is not precisely formulated and fully supported by the results. Namely, the title suggest the general mechanism, while experiments were performed on U-2 OS cell line with epithelial morphology that was derived in 1964 from a moderately differentiated sarcoma of the tibia of a 15-year-old, white, female osteosarcoma patient. The sex-different and age-different responses are very well-known. Accordingly, it is required to use both, primary as well as immortalized cell lines from both sexes to keep title. Also, the age of cells should be precisely stated.
1.2. Please correct “ploymerized” to “polymerized”.
(2) ABSTRACT
2.1. Please describe the results obtained from the new experiments.
2.2. Please consider to “rewrite” the abstract since some of the parts are already reported (“suggesting the importance of PI4P in the division process”; https://pubmed.ncbi.nlm.nih.gov/?term=PI4P+division).
2.3. Please correct “ploymerized” to “polymerized”.
(3) INTRODUCTION
3.1. Please cite the original (instead of reviews) pioneered works as well as recent advances in the field with focus on the subject of the study. It will make your manuscript more attractive to the readers.
3.2. Please correct “ploymerized” to “polymerized”.
(4) MATERIALS AND METHODS
4.1. Please perform new experiments using the use both, primary as well as immortalized cell lines from both sexes. Namely, the experiments were performed on very specific U-2 OS cell line with epithelial morphology that was derived in 1964 from a moderately differentiated sarcoma of the tibia of a 15-year-old, white, female osteosarcoma patient. The sex-different and age-different responses are very well-known. Accordingly, the experiments are required for future consideration.
4.2. Please provide intra- as well as inter-assay coefficients for all analyses.
(5) RESULTS
5.1. Please provide results and figures obtained from the new experiments.
5.2. Please correct “ploymerized” to “polymerized”.
(6) DISCUSSION
6.1. Please discuss the original (not reviews), and important pioneered results, as well as recent advance in the field focusing on the subject of the study.
6.2. Please discuss the new results.
6.3. Please correct “ploymerized” to “polymerized”.
(7) REFERENCES
7.1. Please provide references describing the original (not reviews), and important and pioneered results, but also references describing the recent advance in the field.
(8) FIGURES and FIGURE LEGENDS
8.1. Please provide new figures and figure legends showing the new results.
8.2. Please correct “ploymerized” to “polymerized”.
(9) GENERAL
9.1. Please correct “ploymerized” to “polymerized” through the text, starting from the title.
9.2. Please use, at least “spelling checker” or “Grammarly-on-line” to correct text.
Good luck and all the best!
Reviewer 2 Report
The authors use super-resolution imaging to show the role PI4P vesicles have on mitochondrial division in U-2OS cells. The manuscript is nicely presented and the results support the authors’ conclusion that alteration of PI4P levels alters mitochondrial morphology, PI4P vesicles move throughout the cell via microtubules and ER, the presence of PI4P vesicles stimulates actin polymerization, and finally PI4P and actin polymerization are present just before mitochondrial division in many cases. I only have minor concerns as noted here and on the manuscript itself (attached).
1) Suggest changing wording throughout- longer mitochondria does not directly mean less mitochondrial division. It could also be more fusion, or vice versa. Though it is most an alteration of division rates, based on the localization of PI4P vesicles, fusion events were not looked at and fission was not quantified. A more general conclusion of, dynamics are altered would be more appropriate. See Line 95, for sure.
2)Figure 2H Long distance – It is really hard to follow the mitochondria. Can red signal be made brighter so one can see the mitochondria?
3) Figure 2H Short distance 5s – This image has a 4s label in the center of it.
4) Section 2.2. Conclusion is valid, but I do wonder if perhaps there is something else going on. To fully support this conclusion inhibition of microtubules would be helpful. I don’t think this is a required additional experiment but it would help solidify that microtubules are indeed the main structure responsible.
5) In Figure 3 and 4 it is very hard to see the actin polymerization events, even though they are marked by arrows, this lessens the conclusion trying to be drawn. I am not sure if anything can be done. Figure 3e is very clear with the magnification.
6) Figure 4b the colors of the labels (EGFP-lifeact, mCherry, Mitotracker) above the panel do not match with the actual colors of actin, PI4P, and mitochondria. The figure caption is accurate.
7) My biggest concern and it really isn’t that big, in the discussion please add in some information/theories about why PI4P vesicles might be needed at sites of division. My thoughts- if you are pinching off you may need additional lipids to fill in and seal the hole that is created. I have always wondered where the lipid come from and this may be one method.

Author Response
We thank the Editor for the opportunity to re-submit this manuscript and the reviewers for their very positive and constructive comments. Point-by-point responses to these comments are provided below. Changes made to the manuscript are summarized in our response to the relevant comment. In the revised manuscript, altered/inserted words have been highlighted in blue color. (In addition to these changes, a number of changes to the wording and grammar have been made to improve the readability of the manuscript. These have not necessarily been highlighted, to avoid unnecessary clutter in the revision, but any substantive changes have been marked.) All numerical references to figures, tables, or citations in the manuscript or supplementary information refer to the numbering in the newly submitted version.
Reviewer #2:
- Suggest changing wording throughout- longer mitochondria does not directly mean less mitochondrial division. It could also be more fusion, or vice versa. Though it is most an alteration of division rates, based on the localization of PI4P vesicles, fusion events were not looked at and fission was not quantified. A more general conclusion of, dynamics are altered would be more appropriate. See Line 95, for sure.
Response 1: We have modified the sentences and replaced “and the mitochondrial division rate increased” with “and the mitochondrial dynamics were altered” to illustrate the results more accurate (Page 2).
- Figure 2H Long distance It is really hard to follow the mitochondria. Can red signal be made brighter so one can see the mitochondria?
Response 2: Thanks for the reviewer's suggestion. To address this issue, we have brightened the red signal and updated Figure 2h.
- Figure 2H Short distance 5s – This image has a 4s label in the center of it.
Response 3: We have deleted the extra “4s” label.
- Section 2.2. Conclusion is valid, but I do wonder if perhaps there is something else going on. To fully support this conclusion inhibition of microtubules would be helpful. I don’t think this is a required additional experiment but it would help solidify that microtubules are indeed the main structure responsible.
Response 4: We thank the reviewer for the positive comment.
- In Figure 3 and 4 it is very hard to see the actin polymerization events, even though they are marked by arrows, this lessens the conclusion trying to be drawn. I am not sure if anything can be done. Figure 3e is very clear with the magnification.
Response 5: To address this issue, we have performed additional imaging deconvolution to improve the signal-to-noise ratio and also enlarged the area of polymerization. We have updated Figures 3g and 4c in the revised manuscript to provide clearer views.
- Figure 4b the colors of the labels (EGFP-lifeact, mCherry, Mitotracker) above the panel do not match with the actual colors of actin, PI4P, and mitochondria. The figure caption is accurate.
Response 6: We have updated Figure 4b to match the colors of the labels at the top of the panel with the actual colors.
- My biggest concern and it really isn’t that big, in the discussion please add in some information/theories about why PI4P vesicles might be needed at sites of division. My thoughts if you are pinching off you may need additional lipids to fill in and seal the hole that is created. I have always wondered where the lipid come from and this may be one method.
Response 7: To address this issue, we have added following sentences to discuss why PI4P vesicles might be needed for mitochondrial division. “We found that PI4P-containing vesicles from Golgi can act as a signaling molecule to recruit actin to mitochondrial fission sites, and then actin polymerized at PI4P-associated mitochondrial division sites. Previous reports showed that the forces generated by the actin skeleton can result in cellular membrane deformation and scission30,31,32, and the tension generated by actin polymerization also causes mitochondrial inner and outer membranes to break down21. Our experimental results suggested that PI4P vesicles were not the ultimate mitochondrial division regulator as mentioned in previous studies17, and that it may be the force generated by subsequent actin polymerization that led to the final division of mitochondria.” (Page 13-14)
Round 2
Reviewer 1 Report
IJMS
COMMENTS TO THE EDITORS AND THE AUTHORS
ijms-2288595R1: “PI4P-containing vesicles from Golgi contribute to mitochondrial division by coordinating with polymerized actin”
Dear the Editors and the Authors,
The manuscript is significantly approved.
I do not have additional comments.
All the best ?